# Body Composition in Individuals with Obesity According to Age and Sex: A Cross-Sectional Study

**DOI:** 10.3390/jcm9041188

**Published:** 2020-04-21

**Authors:** Laurent Maïmoun, Thibault Mura, Antoine Avignon, Denis Mariano-Goulart, Ariane Sultan

**Affiliations:** 1Département de Médecine Nucléaire, CHRU Montpellier, 34090 Montpellier, France; d-mariano_goulart@chu-montpellier.fr; 2U1046 INSERM, UMR9214 CNRS, Physiologie et Médecine Expérimentale du Cœur et des Muscles (PHYMEDEX), University of Montpellier, 34295 Montpellier, France; a-avignon@chu-montpellier.fr (A.A.); a-sultan@chu-montpellier.fr (A.S.); 3Département d’Information Médicale, CHRU Nîmes, 30090 Nîmes, France; Thibault.MURA@chu-nimes.fr; 4Département Endocrinologie, Nutrition, Diabète, Equipe Nutrition, Diabète, CHRU Montpellier, 34090 Montpellier, France

**Keywords:** obesity, body composition, sarcopenia, skeletal muscle index, lean tissue mass and fat mass

## Abstract

Obesity is characterized by an alteration in body composition (BC); however, it is not known whether this alteration is modified by aging or sex. The aims of this study were to analyze BC in individuals with obesity based on age and sex and to determine the prevalence of sarcopenia. Seven hundred and fifty-five obese individuals were subdivided into four age groups. The hole (WB) and segmental BC were determined using dual-energy X-ray absorptiometry (DXA). In men, the WB lean tissue mass (LTM) and fat mass (FM) adjusted by weight and height were relatively constant with age. In women, the WBLTM was higher and WBFM was lower in the >65 group compared to the 18–34 group. A decrease in the LTM and FM at lower limbs and an increase in the trunk were observed, particularly in women, inducing a lower appendicular lean mass index (ALMI; appendicular LTM/height^2^) in the >65 group compared to the 18–34 group in both sexes. This study demonstrated that even though the WBLTM and FM are relatively constant with age, individuals with obesity present a localized redistribution of these two components. This body composition change leads to a decrease of the ALMI with age, one of the criteria included in the sarcopenia definition.

## 1. Introduction

Body composition changes with aging have been characterized by a decrease in muscle mass and a relative increase in fat content in normal-weight individuals [1]. These age-related changes are associated with an increased risk of comorbidities, such as type 2 diabetes. Furthermore, a decrease in muscle mass can lead to sarcopenia, a syndrome characterized by the progressive and generalized loss of skeletal muscle mass with aging and weakness or poor physical performance [2]. Sarcopenia has various adverse outcomes such as physical disability, poor quality of life, and an increased risk of falls and fractures, which all heighten the mortality risk [3,4].

Obesity is defined as an excessive fat mass (FM) as a percentage of the body weight, with a relative increase in the lean tissue mass (LTM) [5]. This disease is now common and is also associated with comorbidities [6]. Moreover, it lowers the quality of life and can exacerbate age-related functional decline [7]. The combination of age and obesity, which Baumgartner et al. [8] defined as sarcopenic obesity, may accentuate the disability induced by one of the conditions alone. Although sarcopenic obesity has gained significant attention from the scientific community in recent years, its true prevalence is unclear. This may be because no definition has been universally accepted, as both sarcopenia and obesity have been defined in many ways using different cut-offs [9]. For example, when only the skeletal muscle mass was used to define sarcopenic obesity, its prevalence in obese people 60 years and older ranged by a factor of 19 (from 4.4% to 84%) for men and 26 (from 3.6% to 94%) for women when eight separate research definitions were applied to a representative National Health and Nutrition Examination Survey (NHANES) sample of noninstitutionalized individuals [10]. Several factors may explain the wide variability in these rates, and Batsis et al. [10] specifically noted that the following all varied considerably between studies: thresholds for obesity and sarcopenia, the determination of sarcopenia using appendicular versus total body skeletal mass, sex-specific cut-offs, and ethnicity. This high degree of variability indicates the need for consensus criteria that can be reliably applied across clinical and research settings [11]. Moreover, the lack of consensus surrounding the standard diagnostic criteria for sarcopenic obesity is a serious impediment to determining at-risk patients, the effectiveness of clinical treatments, and the optimal clinical protocols for use in routine care settings, which are critical issues for the healthcare practitioners who must detect and manage this condition [12]. 

Until recently, the thresholds used to determine sarcopenia in individuals with obesity were calculated from the data of non-obese individuals [13]. However, it is quite probable that the population with obesity presents a specific body composition change with aging that has never been directly evaluated, to our knowledge. Quantifying the muscle mass loss according to age in individuals with obesity might therefore enable the calculation of new thresholds specifically adapted for this population.

The aims of this study were thus to (1) assess the whole and segmental body composition in relation to age and sex in individuals with obesity, (2) determine whether the repartition of LTM and FM differs according to age and sex, and (3) determine the prevalence of sarcopenia with age.

## 2. Participants and Methods

Individuals with obesity, defined by a body mass index (BMI) higher than 30 kg/m^2^, were recruited consecutively between December 2012 and September 2016 in the Nutrition Clinic of the University Hospital of Montpellier, France, where they had been referred for a metabolic and physical assessment of their obesity. All the investigations and measurements were performed in fasting conditions in the morning (8:30–10 a.m.). The majority of the participants had a long-standing history of obesity (more than 5 years), and none of them had undergone bariatric surgery. The exclusion criteria were pregnancy, acute medical treatment, and any physical handicap (amputation, neurological lesion, orthopedic prosthesis) that might interfere with the body composition measurement. Moreover, participants with a body weight > 190 kg or height ≥ 192.5 cm were also excluded due to the limitations of the densitometry device. The medical history and menopausal status, when relevant, were obtained by questionnaire. The histories of the smoking status and diabetes mellitus, as well as current medications, were also recorded. The standing height was measured with a stadiometer to the nearest 0.1 cm. The height and weight were measured with participants wearing light clothing and no shoes; the BMI was calculated as the weight in kg divided by the square of the height in meters (kg/m^2^). The waist circumference was recorded to the nearest 0.1 cm midway between the last rib and the crest of the ileum using a non-stretch tape measure. Physical activity levels were not specifically determined. Nevertheless, the individual with obesity only performed leisure physical activities for less than one hour per week. Moreover, none of them were included in a training program at the day of inclusion.

### 2.1. Definition of Comorbidities

Comorbidities were defined according to the usual definitions:

Type 2 diabetes was defined as HbA1c ≥ 6.5%, and/or fasting glycaemia ≥ 7 mmol/L, and/or antidiabetic treatment [14].

Hypertension was defined as systolic blood pressure > 140, and/or diastolic blood pressure > 90, and/or use of anti-hypertensive medications [15].

### 2.2. Participant Consent

All participants gave written informed consent. The study was performed according to the principles of the Declaration of Helsinki and was approved by the local ethics committee (CPP Sud-Méditerranée IV, Montpellier, France). All participants were entered into a registry with the data collected during their hospitalization, including anthropometric, clinical, biological information and body composition determination (NDC-2009-1052).

### 2.3. Body Composition Determination

The procedure was previously described in detail [16]. The soft tissue body composition (FM, kg; percentage of body fat mass, % FM; and LTM, kg) was measured using DXA (Hologic QDR-4500A, Hologic, Inc., Waltham, MA, USA). Data at each localized site (upper limbs, trunk, and lower limbs) were derived from the whole-body scan. The trunk was defined as the whole-body excluding limbs and the head. All scanning and analyses were performed by the same operator to ensure consistency, after following standard quality control procedures. The quality control for DXA was checked daily by scanning a lumbar spine phantom consisting of calcium hydroxyapatite embedded in a cube of thermoplastic resin (DPA/QDR-1; Hologic x-calibre anthropometrical spine phantom). The CVs given by the manufacturer were < 1% for LTM and FM. 

The localized repartition was calculated as follows:Percentage LTM of the upper limbs: the sum of LTM of each upper limb/LTM of whole body × 100.Percentage LTM of the lower limbs: the sum of LTM of each lower limb/LTM of whole body × 100.Percentage LTM of the trunk: LTM of trunk/LTM of whole body × 100.Percentage FM of the upper limbs: the sum of FM of each upper limb/FM of whole body × 100.Percentage FM of the lower limbs: the sum of FM of each lower limb/FM of whole body × 100.Percentage FM of the trunk: FM of trunk/FM of whole body × 100.

### 2.4. Definition of Sarcopenia

The appendicular lean mass (ALM; kg) was defined as the sum of the lean soft tissue mass for the arms and legs, as described by Heymfield et al. [17]. The appendicular lean mass index (ALMI; kg/m^2^) was defined as ALM/height^2^. As ALMI thresholds for the definition of sarcopenia may vary between ethnic groups, like Asian and Caucasian [18,19], we chose to use EWGSOP2 cut-offs (ALMI < 7.0 kg/m^2^ for men and < 5.5 kg/m^2^ in women) that are the most widely accepted definition for sarcopenia [20].

### 2.5. Statistical Analysis 

The characteristics of the participants are described with the proportions for categorical variables and with the means ± standard deviations (SD) for quantitative variables. We analyzed the four groups (18–34; 35–49, 50–64, >65) using two strategies. First, we compared all the age groups over 35 to the 18–34 age group, which was considered as the reference group. Second, we compared all the age groups over 35 years to the adjacent younger age group. For the crude two-by-two comparisons, we used the Student’s *t*-test for quantitative variables and the Chi-square or Fisher’s exact test for qualitative variables. For both sexes, a generational effect was suspected given the significant difference in weight and height between the groups. As these two covariables were potential confounding factors for the LTM and FM values, LTM and FM were adjusted on height and weight and compared between groups using multivariate linear regression models. 

We took into account the multiple testing problem when comparing the age groups two by two, using the Bonferroni (Dunn) procedure in the raw analyses and the Tukey-Kramer procedure in the adjusted analyses.

All analyses were conducted by the Medical Statistics Department of the Montpellier University Hospital using SAS software (V.9.3, SAS Institute, Cary, NC, USA, and R V.3.2.3). A two-sided *p* value of < 0.05 was considered to indicate statistical significance. 

## 3. Results

### 3.1. Participants Characteristics

A total of 755 participants with obesity (549 women (72.7%) and 206 men (27.8%)) with ages ranging from 18.1 to 81.9 years were included. In women, 253 (46.1%) were premenopausal and 296 (53.9%) were postmenopausal. Men and women had a mean BMI of 37.5 ± 4.6 and 38.6 ± 5.4 kg/m^2^, respectively (Table 1). The number and the percentage of participants by BMI class (30–35, 35–40, and > 40 kg/m^2^) were for men (78 (37.9); 72 (34.9); 56 (27.2)) and for women (136 (24.8), 226 (41.2), 187 (34)). The baseline anthropometric characteristics regarding age group and sex are summarized in Table 1. In men, the weight, BMI, and waist circumference were lower in older groups compared to the reference group: 18–34 years. In women, the height was significantly lower and the waist circumference was significantly higher in the 50–64 and > 65 groups than in the reference group. The weight was significantly lower only in the > 65 group in comparison with the 18-34 group. In both sexes, the prevalence of hypertension and diabetes was higher in participants ≥ 50 years.

### 3.2. Body Composition and Appendicular Skeletal Mass 

The adjusted WB and localized LTM and FM for men and women are presented, respectively, in Table 2 and Table 3. In men, the adjusted WB LTM and FM were significantly different only in the 35–49 group compared to the reference group. When site composition analyses were performed, only the trunk LTM was higher and the FM was lower in the oldest group compared to the reference group. In most of the localized sites, the highest value for the LTM and consequently for the ALM was observed in the 35–49 group compared to the reference group. In women, the WB LTM was higher in the > 65 group than in the reference group. The trunk LTM was higher and the lower limbs LTM was lower in all age groups compared to the reference group. This LTM loss resulted in lower ALM values in all age groups compared to the reference group. Concerning the FM, the WB FM was lower only in the > 65 group compared to the reference group. The trunk FM was higher in the 50–64 and > 65 groups compared to the reference group, whereas the lower limbs FM was lower in all groups compared to the reference group.

### 3.3. Repartition of Lean Tissue Mass and Fat Mass

The repartition of LTM and FM is presented in Figure 1. In both men (Figure 1A,B), the % of leg LTM was lower and the % of trunk LTM was higher in the 50–64 and > 65 groups compared to the 18–34 group. In women (Figure 1C,D), the same observation concerning the % of leg LTM was observed, although the difference appeared first in the younger 34–49 group. Concomitantly, in both men and women, the % of leg FM was lower and the % of trunk FM was higher in all groups in comparison with the 18-34 reference group. For men and women, both % LTM and % FM were relatively stable in the upper limbs.

### 3.4. Appendicular Lean Tissue Mass Index and Prevalence of Sarcopenia

The Student’s t-test following by Bonferroni Dunn corrections were used to compare ALMI between age groups. In men, ALMI (kg/m^2^) was significantly different between age groups (*p* < 0.001), and the values were: 10.57 ± 1.02 for 18–34 years; 10.84 ± 1.03 for 35–49 years; 9.87 ± 1.05 for 50–64 years; and 9.73 ± 1.07 for > 65 years, respectively. The ALMI values peaked in the 35–49 group and decreased significantly after 50 years. Then, participants in the 50–64 and > 65 groups presented significantly lower values than those in the 18–34 group, with a mean difference of −6.6% and −7.9%, respectively. In women, ALMI (kg/m^2^) was significantly different between age groups (*p* = 0.005), and the values were: 8.93 ± 1.15 for 18–34 years; 8.76 ± 1.1 for 35–49 years; 8.60 ± 1.05 for 50–64 years; and 8.48 ± 1.14 for > 65 years, respectively. Moreover, ALMI was significantly lower only in the oldest group > 65 years, compared to the youngest group (18–34), with a mean difference of −5.1%. Scatter plots (Figure 2) represent the ALMI according to age in men and women. 

The prevalence of sarcopenia was estimated according to the definition retained for a Caucasian population [20]. Our results indicated that no man or woman with obesity presented sarcopenia. In men, the extreme values for ALMI were 8.8–13.9 kg/m^2^ for individuals aged 34–49 years; 7.5–13.9 kg/m^2^ for individuals aged 50–64 years; and 8–12.5 kg/m^2^ for individuals aged > 65 years. In women, the extreme values for ALMI were 6.7–12.6 kg/m^2^ for 34–49 years; 6.4–12.4 kg/m^2^ for 50–64 years; and 6.7–11.9 kg/m^2^ for > 65 years. 

### 3.5. Relationship Between Type 2 Diabetes and Sarcopenia

As participants with obesity may develop other diseases, particularly type 2 diabetes, which may have an effect on the body composition, we compared ALMI values according to the type 2 diabetes status. However, as the prevalence of diabetes increases with age in individuals with obesity (Table 1) and as weight and height were different between the age groups, the ALMI values were adjusted on these three covariables. The adjusted ALMI was comparable between men with or without diabetes (10.28 ±0.09 kg/m^2^ vs. 10.11 ± 0.09 kg/m^2^), as well as between women (8.77 ± 0.05 vs. 8.68 ± 0.04 kg/m^2^).

## 4. Discussion

The body composition change with aging follows a well-known model and is characterized by a decrease in muscle mass and an increase in fat mass. This has been largely documented in normal-weight individuals [21,22,23]. However, to our knowledge, there are no data on the body composition change with aging in the population with obesity. Only, Tian et al. [24] tried, from NHANES data, to model the age-related changes in the segmental body composition (SBC) according to the BMI (ranging from normal weight to obese). However, these authors assumed that there was a constant BMI-related difference in SBC in all age classes, and thus all of the BMI categories shared the same trends in aging. Conversely, our study included only a large group of both male and female participants with obesity with a wide range of ages (18.1 to 81.9 years), and the body composition was evaluated using the gold standard method (DXA). We showed a body composition change with aging that was different from that observed in non-obese individuals. Indeed, we highlight not only the lack of a significant decrease in WB LTM but also the lack of a significant increase in WB FM. On the other hand, we clearly demonstrate that a localized redistribution of LTM and FM occurs with aging in individuals with obesity. 

### 4.1. Lean Tissue Mass 

Our results show that WB LTM was relatively stable with aging in participants with obesity of both sexes. These data confirm those of Coin et al. [25], who reported the lowest values for WB LTM only after 70 years in normal-weight individuals from geographically similar Mediterranean Caucasian populations. LTM and FM were adjusted on weight and height in this study to better take into account the generational anthropometric characteristics, i.e., declines in weight and height with age that are likely due to lifestyle, eating habits, and physical activity levels [23]. However, these adjustments did not deeply alter the non-adjusted results (data not shown). It is also interesting to note that WB LTM peaked for both sexes in the 35–49 group, although the subsequent lack of variation in WB LTM masked a localized variation in LTM characterized by a progressive reduction in LTM with age at appendicular sites (i.e., lower limbs)—particularly marked in women—and a progressive increase at the central body (i.e., trunk) in both sexes. The reduction in LTM in the lower limbs, ranging from 10% to 14% in the oldest group compared to the youngest group, associated with the lack of variation in the upper limbs, resulted in a lower ALM. The greater percentage decrease in ALM compared to whole-body LTM suggests that the loss of skeletal muscle mass with aging is greater than the loss of non-skeletal muscle mass in the organs of the trunk region [21]. More specifically, the increase in LTM at the trunk in women with obesity may be due to an increased organ size, as the LTM determined by DXA includes the mass of all visceral organs and muscles (smooth and skeletal), bones, ligaments, and tendons, but does not include the FM in internal organs.

The most pertinent criterion to evaluate muscle loss independently of the age-induced height reduction is the ALMI. A reduction in the ALMI, observed earlier from 50–64 years and more markedly after 65 years in men than in women, confirmed the appendicular lean mass decrease with age. Our data agree with those observed in healthy non-obese participants between 18 and 94 years, although the precise age at which sarcopenia starts has remained subject to debate [21,26]. The specific profile of weight and height variations, as well as the body composition according to sex, may explain why the results are not fully comparable between men and women. The result showing the ALMI decreasing with age is of interest because the age-related loss of muscle mass is prevalent in the non-obese elderly and is strongly associated with impaired mobility, increased morbidity and mortality, and a lower quality of life [13,27]. This muscle loss might have even greater effects on the physical functions of participants with obesity. For example, previous studies have shown that climbing stairs, descending stairs, and rising from a chair or bed were significantly more difficult for a sarcopenic obesity group than for the reference group [28,29].

Although we found that the ALMI decreased with age in male and female participants with obesity, none of participants were classified as sarcopenic when conventional definitions for non-obese participants were applied [20]. However, some studies have suggested that sarcopenia is quite prevalent, affecting about 30% of the general population over 65 years [30,31]. This observation suggests two hypotheses: (i) our participants did not suffer from sarcopenia, or (ii) thresholds for sarcopenia defined for normal-weight individuals are inappropriate for participants with obesity. The reduction with age of the lower limbs LTM, ALM, and ALMI points more to the second hypothesis and underscores the crucial need for the definition of new thresholds for sarcopenic obesity specifically adapted to the population with obesity. Recently, in a review, Batsis and Villareal [32] stated that the lack of a consistent definition of sarcopenic obesity constitutes the most notable barrier to advancing the scientific targeting of this condition. Nevertheless, we observed a wide range of ALMI in men and women. This suggests that factors potentially including nutritional (proteins) status, physical activity, drugs, or other factors may influence the ALMI in individuals with obesity. Their identification should be the subject of future studies.

### 4.2. Fat Mass

The adjusted WB FM tended to be relatively constant with age, and the values in the oldest group (>65 years) were lower than in the reference group (18–34 years) in both sexes, but particularly in women. This model of WB FM variation in participants with obesity seems relatively different from the model described in the normal-weight population, which is characterized by an increase in FM, body weight, and BMI up to 70 years, suggesting that the rise in body weight is due to an increase in adipose tissue [25,33,34]. Nevertheless, although WB FM was relatively constant with age in our population with obesity, we found, as observed for LTM, a localized redistribution of FM, characterized by a reduction in FM in the lower limbs and an increase in the trunk in both sexes, but more particularly in women. It is widely acknowledged that menopause increases body weight and FM, especially in visceral areas [35], and that this shift in fat deposition to the center of the body expands the waist circumference [36], as observed in our female participants with obesity. Our results appear similar to those of previous studies of normal-weight individuals showing FM accumulation in absolute terms and as a percentage of the body weight after middle age and a redistribution more to the abdominal region than to the peripheral adipose tissues [21,22,23]. This localized truncal fat gain may explain the increase in the prevalence of type 2 diabetes and hypertension observed in our study.

Schematically, older individuals with obesity may have a profile that Batsis and Villareal [32] described as “fat frail”.

Sarcopenia and obesity have numerous consequences for metabolic parameters, comorbidities, mortality, and the quality of life. For example, they have been associated with insulin resistance [37] and a higher risk of falling [38]. In our study, although we found an increase in type 2 diabetes prevalence with age in both the male and female participants with obesity, we did not find a significant difference in ALMI values between those with and without diabetes. These results may underscore the accuracy of the ALMI measurement in determining the LM variation with the aging process, even though it is not useful for reflecting the expected metabolic profiles. This may be explained by the fact that the ALMI does not include other determinants’ parameters, such as weight size or visceral adipose tissue, which are two independent risk factors for hypertension and type 2 diabetes.

### 4.3. Limitations of the Study

The main limitations of this study are the cross-sectional design and, as underlined by Coin et al. [23], its inherent bias due to generational effects that may have had an impact on the body composition, weight, and height variations. Consequently, the results from this study should be interpreted with caution and may not be generalized to other groups with different BMI values or ethnicities. Particularly, comparing individuals with obesity from different age groups should not reflect what we see during the weight gain process. However, these limitations are mitigated by the weight and height adjustments that did not deeply modify the results, the large number of participants of both sexes included in each age group, and the use of the reference technique (DXA) for the precise evaluation of the whole-body composition and particularly the appendicular LTM, which is recommended for sarcopenia diagnosis [2]. We could not further assume that the duration of obesity was comparable within the subject age groups. We are aware that the quantification of LTM is only one part of the definition of sarcopenia and that it needs to be completed in the future by the exploration of muscle strength, physical performance, and/or physical function, which may be assessed by one of the following tests: hand grip strength or knee flexion/extension. A recent study demonstrated, however, that a limited variation in the cut-off point for the muscle mass had more impact than changes in cut-off points for gait speed and for grip strength on sarcopenia prevalence [39], although another study demonstrated that muscle strength is a stronger predictor of long-term functional decline than muscle mass [40]. Further studies are needed to evaluate whether body composition changes induce a functional limitation specifically in the population with obesity.

## 5. Conclusions

In conclusion, this study highlighted that individuals with obesity presented with age a localized redistribution of LTM and FM characterized by a truncal accumulation of these two components and a decrease in lower limbs LTM. Despite the ALMI decreasing with age, no individual with obesity presented sarcopenia, suggesting that the thresholds currently used are not adapted to this population.

## Figures and Tables

**Figure 1 jcm-09-01188-f001:**
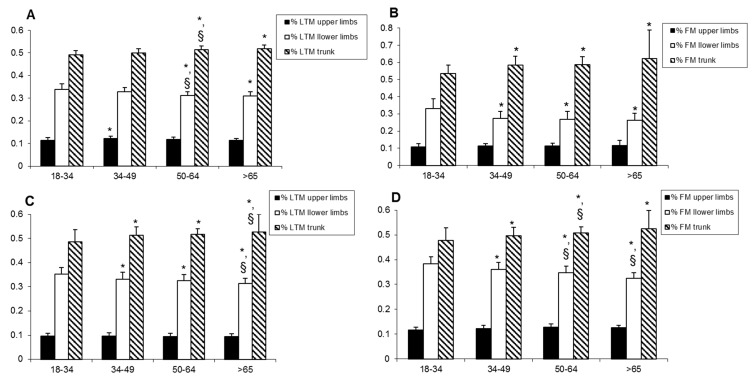
Repartition of lean tissue mass and fat mass in (**A**,**B**) male and (**C**,**D**) female obese participants according to age. Data are given by the mean ± SD. LTM: lean tissue mass; FM: fat mass. The Student’s *t*-test and Bonferroni Dunn procedure to control for the multiple comparisons problem were used to compare age groups. * indicates a significant difference with the 18–34 group, ^§^ indicates a significant difference with the adjacent younger age group.

**Figure 2 jcm-09-01188-f002:**
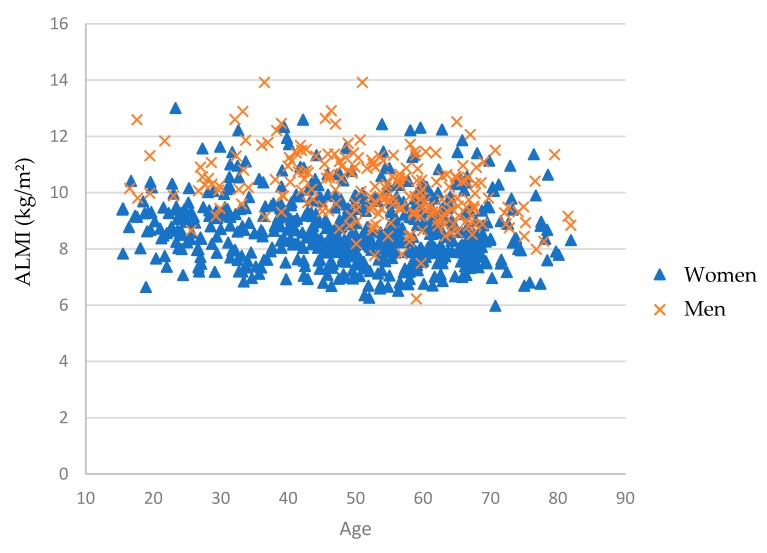
Scatter plots of the appendicular lean mass index (ALMI) in men (ˣ) and women (Δ) obese participants according to age.

**Table 1 jcm-09-01188-t001:** Characteristics of the study population according to sex and age bracket.

	All Participants	18–34 Years	35–49 Years	50–64 Years	>65 Years
**Men**					
Number of participants	206	26	53	88	39
Age (years)	52.6 ± 13.4	28.6 ± 4.6	43.8 ± 3.9	57.3 ± 4.2	70.1 ± 4.9
Weight (kg)	114.7 ± 15.9	123.1 ± 16.4	119.4 ± 16.9	112.2 ± 14.3 *	108.5 ± 14.2 *
Height (cm)	175.0 ± 6.9	173.3 ± 7.5	177.7 ± 6.1 *	175.2 ± 6.8	172.0 ± 6.2
BMI (kg/m^2^)	37.5 ± 4.6	41.0 ± 5.0	37.7 ± 4.6 *	36.6 ± 4.1 *	36.7 ± 4.6 *
Waist circumference (cm)	121.9 ± 11.8	123.4 ± 9.8	122.0 ± 11.5 *	121.4 ± 13.1 *	121.9 ± 10.3 *
Hip circumference (cm)	117.4 ± 11.7	124.3 ± 9.8	116.8 ± 12.2	115.9 ± 11.8	116.3 ± 11.0
WB FM (%)	35.3 ± 5.9	40.1 ± 5.0	34.3 ± 6.0 *	34.7 ± 5.8 *	35.1 ± 5.3 *
WB FM (kg)	41.4 ± 10.9	49.3 ± 10.0	42.2 ± 11.8 *	39.7 ± 10.1 *	38.9 ± 9.7 *
WB LTM (kg)	71.6 ± 8.1	70.2 ± 7.0	76.3 ± 8.0 *	70.9 ± 7.8 §	68.0 ± 7.2
HTA (number; %)	129 (62,6%)	4 (15.4%)	28 (52.8%) *	65 (73.9%) *,^§^	32 (82.1%) *
Diabetes (number; %)	94 (45.6%)	6 (23.1%)	23 (43.4%)	56 (63.6%) *,^§^	28 (71.8%) *
**Smoking (*n*, %)** (current, former, never)	70 (36.3), 38 (19.7), 85 (44.0)	14 (58.3), 7 (29.2), 3 (12.5)	21 (42.0), 9 (18.0), 20 (40)	26 (31.7), 16 (19.5), 40 (48.8) *	9 (24.3), 6 (16.2), 22 (59.5) *
**Women**					
Number of participants	549	109	170	185	85
Age	49.2 ± 14.6	27.0 ± 4.6	44.1 ± 4.2	57.5 ± 4.2	69.7 ± 4.2
Weight (kg)	101.5 ± 15.1	103.2 ± 15.5	102.3 ± 15.2	102.1 ± 14.1	96.2 ± 15.2 *,^§^
Height (cm)	162.01 ± 6.4	164.4 ± 6.2	163.2 ± 5.9	161.5 ± 6.5 *	158.1 ± 5.7 *,^§^
BMI (kg/m^2^)	38.6 ± 5.4	38.2 ± 5.3	38.4 ± 5.2	39.2 ± 5.3	38.4 ± 5.6
Waist circumference (cm)	110.5 ± 12.4	107.4 ± 14.1	109.6 ± 11.6	112.0 ± 11.7 *	112.9 ± 12.0 *
Hip circumference (cm)	124.2 ± 12.1	124.82 ± 11.2	123.9 ± 12.3	125.1 ± 11.9	122.0 ± 13.4
WB FM (%)	45.2 ± 4.5	45.6 ± 4.1	44.7 ± 4.5	46.0 ± 4.3	44.3 ± 5.2 *,^§^
WB FM (kg)	46.7 ± 10.1	47.6 ± 9.4	46.7 ± 10.5	47.6 ± 9.6	43.3 ± 10.5 *,^§^
WB LTM (kg)	53.5 ±6.4	53.8 ± 6.6	54.5 ± 6.5	53.2 ± 6.0	51.8 ± 6.4
HTA (number; %)	221(40.2%)	7 (6.4%)	43 (25.3%) *,^§^	105 (56.8%) *,^§^	66 (77.7%) *,^§^
Diabetes (number; %)	144 (26.2%)	9 (8.3%)	35 (20.6) *,^§^	73 (39.5%) *,^§^	50 (58.9%) *,^§^
**Smoking (*n*, %)** (current, former, never)	284 (60.4), 84 (17.9), 102 (21.7)	65 (58.6), 32 (28.8), 14 (12.6)	105 (53.0), 40 (20.2), 53 (26.8)	123 (51.5), 36 (15.1), 80 (33.4) *	61 (53.0), 14 (12.2), 40 (34.8) *

Data are presented as the mean ± SD, except for HTA and diabetes which are presented by the number and the percentage (%). BMI: body mass index; WB: whole body; LTM: lean tissue mass; FM: fat mass; HTA: hypertension. * denotes a significant difference compared to the 18–34 group. ^§^ denotes a significant difference compared to the adjacent younger age group. The Student’s *t*-test for quantitative variables or the Chi-square test for qualitative variables, and the Bonferroni Dunn procedure to control for the multiple comparisons problem, were used for the statistical analysis.

**Table 2 jcm-09-01188-t002:** Segmental body composition adjusted for weight and height according to the age bracket in men.

Age	18–34 Years	35–49 Years	50–64 Years	>65 Years
Number of participants	26	53	88	39
**Lean tissue mass**				
Whole body (kg)	69.8 ± 0.9	73.4 ± 0.6 *	71.2 ± 0.5 ^§^	71.3 ± 0.7
Trunk (kg)	34.6 ± 0.5	36.7 ± 0.4 *	36.5 ± 0.3 *	36.8 ± 0.4 *
Upper limbs (kg)	6.0 ± 0.2	6.7 ± 0.1 *	6.1 ± 0.9 ^§^	6.0 ± 0.1
Lower limbs (kg)	17.3 ± 0.3	17.8 ± 0.2	16.7 ± 0.2 ^§^	16.7 ± 0.2
ALM (kg)	31.2 ± 0.6	32.9 ± 0.4 *	30.7 ± 0.3 ^§^	30.5 ± 0.47
**Fat mass**				
Whole body (kg)	43.5 ± 1.0	40.1 ± 0.7 *	41.4 ± 0.6	41.6 ± 0.8
Whole body (%)	37.1 ± 0.8	34.0 ± 0.6 *	35.5 ± 0.4	35.6 ± 0.6
Trunk (kg)	23.0 ± 0.8	23.6 ± 0.6	24.2 ± 0.4	25.5 ± 0.6 *
Upper limbs (kg)	3.5 ± 0.2	3.5 ± 0.1	3.5 ± 0.1	3.6 ± 0.1
Lower limbs (kg)	11.0 ± 0.4	8.2 ± 0.3 *	8.5 ± 0.2 *	8.5 ± 0.3 *

Data are presented as the mean ± SEM. ALM: appendicular lean mass. Least squares mean estimate using a multivariate linear regression model, for an obese man of 114.7 kg and 175 cm. We used the Tukey-Kramer procedure to control for the multiple comparison problem. * denotes a significant difference compared to the 18–34 group. ^§^ denotes a significant difference compared to the adjacent younger age group.

**Table 3 jcm-09-01188-t003:** Segmental body composition adjusted for weight and height according to the age bracket in women.

Age	18–34 Years	35–49 Years	50–64 Years	>65 Years
Number of participants	109	170	185	85
**Lean tissue mass**				
Whole body (kg)	52.7 ± 0.3	53.8 ± 0.3 *	53.2 ± 0.3	54.5 ± 0.4 *,^§^
Trunk (kg)	25.6 ± 0.3	27.6 ± 0.3 *	27.6 ± 0.2 *	28.7 ± 0.4 *,^§^
Upper limbs (kg)	3.8 ± 0.1	3.9 ± 0.1	3.8 ± 0.1	3.8 ± 0.6
Lower limbs (kg)	13.8 ± 0.1	13.3 ± 0.1 *	12.9 ± 0.1 *,^§^	12.9 ± 0.1 *
ALM (kg)	23.6 ± 0.2	23.1 ± 0.1 *	22.4 ± 0.1 *,^§^	22.6 ± 0.2 *
**Fat mass**				
Whole body (kg)	47.3 ± 0.3	46.3 ± 0.3	47.1 ± 0.3	45.6 ± 0.4 *,^§^
Whole body (%)	45.9 ± 0.3	44.8 ± 0.3 *	45.7 ± 0.3	44.2 ± 0.4 *
Trunk (kg)	22.7 ± 0.2	23.0 ± 0.2	23.9 ± 0.2 *,^§^	23.7 ± 0.3 *
Upper limbs (kg)	4.2 ± 0.1	4.3 ± 0.1	4.5 ± 0.1	4.2 ± 0.1
Lower limbs (kg)	13.5 ± 0.3	12.5 ± 0.2 *	12.4 ± 0.2 *	11.4 ± 0.3 *,^§^

Data are presented as the mean ± SEM. ALM: appendicular lean mass. Least squares mean estimate using a multivariate linear regression model, for an obese woman of 101.5 kg and 162 cm. We used the Tukey-Kramer procedure to control for the multiple comparison problem. * denotes a significant difference compared to the 18–34 group. ^§^ denotes a significant difference compared to the adjacent younger age group.

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
