# Peer review of "Body Composition in Individuals with Obesity According to Age and Sex: A Cross-Sectional Study"

_jcm, 2020, doi:10.3390/jcm9041188_

Round 1

Reviewer 1 Report

Minor comments:

  1. Figure 2: femme and homme should be changed to women and men
  2. Conclusion should summarize the end result. Discussing other findings should be moved to discussion part

Author Response

Reviewer 1 :

  1. Figure 2: femme and homme should be changed to women and men

Done

  1. Conclusion should summarize the end result. Discussing other findings should be moved to discussion part

Thank you for this comments. The conclusion was modified as follow: “In conclusion, this study highlighted that individual with obesity presented with age a localized redistribution of LTM and FM characterized by a truncal accumulation of these two components and a decrease in lower limbs LTM. Despite ALMI decreased with age, no individual with obesity presented sarcopenia, suggesting that the thresholds currently used is not adapted to this population. “

Moreover, we moved the sentence “Schematically, older individuals with obesity may have a profile that Batsis and Villareal [32] described as “fat frail”” at page 10 line 298-299.

Reviewer 2 Report

Thank you for applying my comments!

Author Response

Reviewer 2: Thank you for applying my comments!

Thank you for this comment

This manuscript is a resubmission of an earlier submission. The following is a list of the peer review reports and author responses from that submission.

Round 1

Reviewer 1 Report

How height was measured?

What was the rational for using t-test when there were four groups? 

Hydration status can affect LTM  was there any data on hydration status of participants?

Reviewer 2 Report

I reviewed the article entitled: " Body Composition in Patients with Obesity According to Age and Gender: A Cross-Sectional Study", by Laurent Maïmoun, et al.

The authors conducted a cross-sectional study to assess body composition in obese participants. They categorized the patients, men and women, into four groups based on their ages. They concluded that the body composition is almost similar in different age groups of obese patients, and then there is not significant age-related changes in body composition of obese patients.

  • The study is a well-written manuscript. The idea of assessing body composition in obese people of different age groups is relatively novel. However, a study discussed body compositions at different age groups and BMIs.

A multivariate model for predicting segmental body composition. Tian S, Mioche L, Denis JB, Morio B. Br J Nutr. 2013 Dec; 110 (12): 2260-70.

  • I think weigh gain can affect whole-body composition, and then changes metabolic profile. As a result, comparing obese people of different age groups should no reflect what we see during the weight gain process.
  • Elderly is associated with increased incidence of age-related diseases such as T2DM, obesity, metabolic syndrome, and cardiovascular diseases. Age-related loss of lean mass and acquisition of fat mass is relatively well studied. In addition, changes in body composition is considered as a marker that can predict the mentioned diseases. I think it is possible to get wrong results in a cross-sectional study. The conclusion that lean mass and fat mass are relatively constant with age may need a cohort study. However, it is reported that the process of weight gain and aging together led to the age-related changes, losing lean mass and gaining fat mass.

Aging and Imaging Assessment of Body Composition: From Fat to Facts. Ponti F, Santoro A, Mercatelli D, Gasperini C, Conte M, Martucci M, Sangiorgi L, Franceschi C, Bazzocchi A. Front Endocrinol (Lausanne). 2020 Jan 14; 10: 861.

Age-Related Changes in Segmental Body Composition by Ethnicity and History of Weight Change across the Adult Lifespan. Tian S, Morio B, Denis JB, Mioche L. Int J Environ Res Public Health. 2016 Aug 13; 13(8). pii: E821.

  • The men older group had lower BMI, which technically affect other body compositions. Then lack of significant change in WB LTM and FM could be due the baselined differences. However, the SMI values are compatible with aging process, but no significant correlation with development of diabetes, which may underscore the accuracy of SMI measurement but not reflecting the expected metabolic profiles.

Minor comments: writing needs edition

  • There are some parts that have double spaces
  • Discussion, second part “lean tissue mass”, paragraph 3rd: “Although we found that SMI decreased with age in male and female patientswith obesity”

Reviewer 3 Report

This manuscript simply summarises data from a convenience sample and reports means+/- SD in an area that is not novel and therefore does not further add to the existing literature. The data really needs to be completely reanalysed and presented, to address an unanswered question.  

  • Pg 1 line 38: Please remove the word “subjects” as this is not acceptable in 2020. Replace with “individuals” or “participants”.
  • Pg 1 line 40: Sarcopenia is not only reduced muscle mass but also muscle strength. Please rephrase to correctly define sarcopenia.
  • Pg 2 line 54: NHABES typo, it should be NHANES
  • Pg 2 line 58: please replace the word gender with sex throughout the manuscript. Sex refers to the biological differences between males and females, while gender is more of a sociological term.
  • Pg 2 line 88: authors should provide references for the definition of the comorbidities.
  • Pg 3 line 119: Appendicular skeletal muscle mass should be changed to Appendicular lean mass (ALM). The DXA does not only measure skeletal muscle, there are the ligaments and other soft tissue that are taken into the measurement. ALM is more acceptable.
  • Pg 3 line 120: skeletal muscle index needs to be changed to Appendicular lean mass index (ALMI) – see previous comment.
  • Pg 3 line 120: reword Asiatic to Asian.
  • Pg 3 line 122: the preference of using the Baumgartner thresholds is not clear. The authors do not mention the EWGSOP2 or the FNIH cut-points at all – and these are the two most widely accepted and used cut-points. Please indicate why neither of these were used. Simply stating “…were similar to our French patients with obesity” does not suffice. This is quite a large area in the field of sarcopenia – using one definition as opposed to another can change the findings of a study.
  • Pg 4 line 142: please do not use the word patient; Participant would be more appropriate. Also please change throughout manuscript.
  • Pg 4/5 Table 1 and 2: the table has NO footnotes- all acronyms need to be expanded in the footnotes, you need to say how the data is expressed ie mean +/- SD, what is in the parentheses is unclear as this also needs to be mentioned in the footnote. The n number can be listed in the same cell as “women” and “men” and does not need its own row. The tables also do not need separate classifications of anthropometry and clinical data as it is redundant in this scenario. Although the stats were explained in the statistical analysis section, you also need to write what stats were performed in the footnote of the table. So the table should be able to be understood as a stand alone table.
  • Tables 1 and 2: these two tables can be merged into one ie have one column for “total” and have the data from table 1 inserted into table 2. Having both is redundant.
  • Pg 5 table 2 and 3: what do all the symbols mean??? Similar to the comments above, you need to have footnotes explaining everything in the table.
  • Pg 5, 6, tables 3 and 4: the titles do not make sense.
  • Figure 2: box plots are not visually the correct graph type for presenting this data. Scatter plots would be far more effective and the peak ALM could be more clearly visualised. In fact, the authors could produce a scatter plot which has both men and women on the same plot.
  • The order of the results do not correspond to the order of the tables. Pg 8 line 203 begins to report data from Table 2 - this is the last results section…